# Combining Medicinal Plant In Vitro Culture with Machine Learning Technologies for Maximizing the Production of Phenolic Compounds

**DOI:** 10.3390/antiox9030210

**Published:** 2020-03-04

**Authors:** Pascual García-Pérez, Eva Lozano-Milo, Mariana Landín, Pedro Pablo Gallego

**Affiliations:** 1Plant Biology and Soil Science Department, Biology Faculty, University of Vigo, 36310 Vigo, Spain; pasgarcia@uvigo.es (P.G.-P.); e.lozanomilo@gmail.com (E.L.-M.); 2Pharmacology, Pharmacy and Pharmaceutical Technology Department, Faculty of Pharmacy, University of Santiago, E-15782 Santiago de Compostela, Spain; m.landin@usc.es

**Keywords:** antioxidants, artificial intelligence, biotechnology, fuzzy logic, *Kalanchoe*, phytochemistry, plant tissue culture, polyphenols, secondary metabolites

## Abstract

We combined machine learning and plant in vitro culture methodologies as a novel approach for unraveling the phytochemical potential of unexploited medicinal plants. In order to induce phenolic compound biosynthesis, the in vitro culture of three different species of *Bryophyllum* under nutritional stress was established. To optimize phenolic extraction, four solvents with different MeOH proportions were used, and total phenolic content (TPC), flavonoid content (FC) and radical-scavenging activity (RSA) were determined. All results were subjected to data modeling with the application of artificial neural networks to provide insight into the significant factors that influence such multifactorial processes. Our findings suggest that aerial parts accumulate a higher proportion of phenolic compounds and flavonoids in comparison to roots. TPC was increased under ammonium concentrations below 15 mM, and their extraction was maximum when using solvents with intermediate methanol proportions (55–85%). The same behavior was reported for RSA, and, conversely, FC was independent of culture media composition, and their extraction was enhanced using solvents with high methanol proportions (>85%). These findings confer a wide perspective about the relationship between abiotic stress and secondary metabolism and could serve as the starting point for the optimization of bioactive compound production at a biotechnological scale.

## 1. Introduction

Medicinal plant research has arisen exponentially in the last few years, thanks to the countless applications developed for plant secondary metabolites, mainly in the fields of drug and food industries, being used as pharmaceuticals and food additives [1]. Among all the different classes of secondary metabolites, phenolic compounds constitute the largest family with more than 8000 compounds identified to date, including many heterogenous subfamilies, such as phenolic acids, flavonoids, anthocyanins and stilbenes, etc. Thus, polyphenols have gained much attention in biotechnology and pharmacology as they possess a wide range of associated bioactivities, acting as antioxidant, anti-inflammatory and anticarcinogenic agents. As antioxidants, polyphenols have been proven to play an efficient role in different processes related to oxidative stress, as in the case of the scavenging of reactive oxygen species (ROS) and derived free-radicals [2].

Being secondary metabolites, phenolics are usually found in limited amounts within plant organisms and their biosynthesis is sensitive to different stress conditions, derived from both biotic, i.e., pathogen-induced damage, and abiotic stresses, i.e., drought, extreme temperatures and nutritional deficiencies [3]. As a result, plant in vitro culture confers a reliable system to promote phenolic accumulation under controlled stress conditions. Nevertheless, the accumulation of phenolic compounds by in vitro-cultured plants depends on a sum of factors that are usually underestimated, such as the mineral composition of culture media, environmental growth conditions, the interaction with endogenous substances and genotype [4]. Additionally, no universal protocol has been achieved for the extraction of phenolics since it is also dependent on a set of factors that influence extraction yields. Phenolic compounds are usually extracted from plant matrices using different aqueous alcoholic solvents that may not encompass an efficient recovery of these compounds in their active form [5]. As a multifactorial phenomenon, the production of phenolic compounds requires the identification of significant factors throughout the construction of large datasets, and their analysis and interpretation may be difficult to achieve.

For such purposes, novel approaches must be developed, as is the case of neurofuzzy logic. This machine learning technology combines artificial neural networks (ANNs) with fuzzy logic to perform the modeling of large and unmanageable databases aimed at identifying significant factors that cause an improvement of a specific response, such as phenolic compound production [6]. Furthermore, the models predicted by neurofuzzy logic are able to facilitate the interpretation of results, by simplifying the detection of optimal responses throughout the formulation of “if-then” rules [7]. The potential from ANN application on the predictability and optimization of multifactorial processes makes this computer-based tool a robust approach to unravel the phytochemical potential of unexploited medicinal plants, as in the case of *Bryophyllum* species.

*Bryophyllum* constitutes a subgenus within the complex *Kalanchoe* genus (Crassulaceae) that contains a number of species widely used in folk medicine across Africa, Asia and South America for the treatment of infections and chronic diseases, such as cardiovascular, neoplastic and inflammatory diseases [8]. Phytochemical analyses performed with several species have shown that bufadienolides and phenolic compounds are the main secondary metabolites responsible for the bioactivities associated with *Bryophyllum* [9,10,11].

In this work, we combine the application of medicinal plant in vitro culture with machine learning algorithms as a primary approach to decipher the key factors that impact the accumulation of secondary metabolites. In this case, different species from subgenus *Bryophyllum* cultured in vitro under abiotic stress will be used as a source of bioactive compounds. Cultures will be subjected to nutritional stress by reducing the concentration of macronutrients, and ultrasound-assisted solvent extraction will be performed to determine the accumulation of phenolic compounds with antioxidant activity. Finally, the information derived from the experimental data will be described according to predictive ANN models for each studied variable, leading to the simplification of result interpretation.

## 2. Materials and Methods

### 2.1. Chemicals

All reagents were of analytical grade. Gallic acid, sodium carbonate, quercetin and 2,2-diphenylpicrylhydrazyl (DPPH) were purchased from Sigma Aldrich (Madrid, Spain). Folin–Ciocalteu’s reagent, aluminum chloride and methanol were purchased from VWR Chemical (Barcelona, Spain). All culture medium reagents were plant tissue culture tested. Milli-Q water was used for phenolic extraction and subsequent determinations.

### 2.2. Plant Material

Three different *Bryophyllum* species were used in this work, namely: *Bryophyllum daigremontianum* Raym.—Hamet et Perr. (BD), *Bryophyllum tubiflorum* Harv (BT) and *Bryophyllum* × *houghtonii* D.B. Ward (*Bryophyllum daigremontianum* × *tubiflorum*, BH). For the establishment of plant in vitro culture, epiphyllous plantlets from 18-month old plants were harvested from a local greenhouse (42°12′40.0″ N 8°43′36.1″ W, Vigo, Spain) in winter 2018 and disinfected according to previous works [5]. Once disinfected, plantlets were placed by groups of three into culture vessels containing 25 mL of sterile culture medium, and four culture vessels were used for each treatment. Two different culture media formulations were used in this work; full strength Murashige and Skoog (MS) medium [12] and MS medium with half-strength macronutrient concentration, named 1/2 MS (Appendix A). Culture media were supplemented with 3% (*w*/*v*) sucrose, solidified with 0.8% (*w*/*v*) agar at pH 5.8, and autoclaved at 121 °C and 1.1 atm for 20 min. Cultures were introduced into growth chambers and subjected to a photoperiod of 16 h light (55 µmol m^−2^ s^−1^) and 8 h dark at 25 ± 1 °C for periodic 12-week subcultures, using epiphyllous plantlets as the explant for successive subcultures.

### 2.3. Extraction of Phenolic Compounds

Plants from the first four subcultures were combined and separated into aerial parts and roots. All samples were frozen at −20 °C, lyophilized and homogenized to get a fine powder. Four different solvents were used to perform phenolic extraction, based on different MeOH:water (*v*/*v*) ratios, i.e., 40% MeOH (M40), 60% MeOH (M60), 80% MeOH (M80) and absolute MeOH (M100). Briefly, 100 mg of dry weight (DW) of plant materials was mixed with 10 mL of each solvent, vortexed, and subjected to solvent extraction at 60 °C in a water bath for 10 min. Samples were then cooled down to room temperature and sonicated for 30 min, prior to their centrifugation at 3500 rpm for 10 min. The supernatants were then separated, cooled down to room temperature, and filtered through 0.45 µm PTFE membrane filters to obtain phenolic extracts. Extracts were stored at 4 °C until use. The extraction procedure was carried out in triplicate.

### 2.4. Total Phenolic Content Determination

Total phenolic content (TPC) determination was carried out following the protocol described by Ainsworth and Gillespie [13]: 100 µL of phenolic extracts were mixed with 200 µL of 10% (*v*/*v*) Folin-Ciocalteu’s reagent and incubated for 2 min at room temperature in the dark. Next, 800 µL of 0.7 M sodium carbonate was added, and the samples were vortexed and incubated for 2 h at 25 ± 1 °C in the dark. The absorbance was measured at λ = 765 nm against a blank and a calibration curve was performed using gallic acid as standard. Results were expressed as gallic acid equivalents (GAE) in mg/g DW. All determinations were carried out in triplicate.

### 2.5. Flavonoid Content Determination

Flavonoid content (FC) determination was carried out based on the protocol described by Pekal and Pyrzynska [14]. One milliliter of phenolic extract was mixed with 0.5 mL 2% (*w*/*v*) aluminum chloride and 0.5 mL of water. Mixtures were vortexed and incubated for 10 min at 25 ± 1 °C in the dark. The absorbance was measured at λ = 425 nm against a blank and a calibration curve was performed using quercetin as standard. Results were expressed as quercetin equivalents (QE) in mg/g DW. All determinations were carried out in triplicate.

### 2.6. Radical Scavenging Activity Determination

The antioxidant activity of phenolic extracts was obtained by the determination of their radical scavenging activity (RSA) against 2,2-diphenylpicrylhydrazyl (DPPH). DPPH is a stable free-radical with purplish coloration, that is quenched under the presence of antioxidants contained in plant extracts. RSA was determined following the procedure developed by Jagtap and coworkers [15]. Briefly, 2850 µL of 110 µM DPPH methanolic solution was mixed with 150 µL of phenolic extracts, and the mixture was vortexed and incubated for 24 h at 25 ± 1 °C in the dark. The decrease in the DPPH signal was measured spectrophotometrically at λ = 517 nm against a blank. Results were expressed as inhibitory concentration 50 (IC50), which expresses the extract concentration (in mg DW mL^−1^) required to reach an inhibition by 50% of the DPPH signal. Additionally, RSA for reference compounds, both gallic acid and quercetin, was also determined under the same conditions, and results were expressed as the percentage of DPPH inhibition as a function of concentration. All determinations were carried out in triplicate.

### 2.7. Statistical Analysis

Continuous data from TPC, FC and RSA were statistically analyzed by one-way analysis of the variance (ANOVA) followed by Tukey’s honest significant difference (HSD) post hoc test at *p*-value *p* < 0.05, using STATISTICA v.12 software (StatSoft Inc., 2014, Street Tulsa, OK, USA).

### 2.8. Modeling Tools

All experimental data were merged into a unique database (Appendix A) and analyzed through FormRules^®^ 4.03 (Intelligensys, Ltd., North Yorkshire, UK) neurofuzzy logic software. For the construction of such a database, macronutrient salts used in culture media were previously expressed into their constitutive ions in order to avoid ion confounding (Table 1) [16].

The experimental design consisted in the combination of 3 different genotypes (BD, BH and BT) × 2 levels of organs (aerial parts and roots) × 2 levels of ion concentration (MS at full and 1/2 strength) × 4 levels of solvents (M40, M60, M80 and M100) accounting for 48 combinations (Appendix A).

The training parameters set for the modeling procedure are shown in Table 2. The structural risk minimization principle (SRM) was the statistical fitness criterion selected since it enables the construction of the best model with minimum generalization error and the simplest rules [17]. After model establishment, submodels were obtained by the application of the adaptative-spline-modeling-of-data mode (ASMOD) performed by FormRules^®^.

The application of neurofuzzy logic confers several advantages facing the simplification of result interpretation since the values related to inputs [18]: (1) were expressed as “if–then” rules; (2) were ranged at different levels (low, medium or high) according to model results; and (3) were combined with a membership degree that takes a certain value between 0 and 1. Additionally, for every output, independent predicted models were provided, and their quality was assessed according to the determination coefficient of the training set, Train Set R^2^, expressed as a percentage obtained from Equation (1):(1)$$R^{2}=\left(1-\frac{\Sigma_{i=1}^{n}{\left(y_{i}-y_{i}^{\prime }\right)}^{2}}{\Sigma_{i=1}^{n}{\left(y_{i}-y_{i}^{\prime \prime }\right)}^{2}}\right)\times 100$$
where $y_{i}$ stands for the experimental value from the dataset, $y_{i}^{\prime }$ stands for the predicted value obtained by the model and $y_{i}^{\prime \prime }$ stands for the mean value of the dependent variable. Significant predictive values were considered according to Train Set R^2^ values between 70–99.9%. Values higher than 99.9% indicate model overfitting, and readjustments would be required [19]. With the aim of assessing model accuracy, ANOVA was performed to check significant differences between experimental and predicted data.

## 3. Results

The results from the determination of total phenolic content (TPC), flavonoid content (FC) and radical scavenging activity (RSA) are shown in Figure 1. Altogether, the results obtained for each parameter depend on different factors that could not be easily identified, even when statistical ANOVA was performed. In this sense, ANOVA could only manage different factors at once, but they present a lack of power for detecting non-linear interactions between different factors. Additionally, the analysis of results led to a difficult interpretation, as it was overwhelmed by data heterogeneity of variables used as factors [20].

Concerning TPC, the results show that the maximum value was obtained by aerial parts from BH cultured in 1/2 MS medium and extracted with M80; 50.0 GAE, mg/g DW (Figure 1A). For all species, the use of 1/2 MS caused a significant increase in the accumulation of phenolic compounds in aerial parts, but the solvent required for the maximum values was different; whereas M80 worked better for BD and BH, M60 was the most efficient solvent for BT (Figure 1A). Conversely, in the case of roots, the highest TPC values were obtained for BD when the solvent was M60, independently on the culture medium used; 19.5 GAE, mg/g DW, for MS and 19.6 GAE mg/g DW, for 1/2 MS (Figure 1B). For BH roots, culture medium showed no influence, as the highest values were obtained with M60 in both cases with no significant differences: 14.0 GAE mg/g DW, for MS and 14.9 GAE mg/g DW, for 1/2 MS; however, solvent influence was significant in the case of 1/2 MS, as M40, M60 and M80 showed similar results. In the case of BT roots, in contrast, phenolic accumulation increased significantly when using 1/2 MS as the culture medium and both M60 and M80 solvents; 15.0 GAE, mg/g DW, and 13.3 GAE, mg/g DW, respectively (Figure 1B).

For FC, results showed a clear behavior in aerial parts, being M100 the most efficient solvent in all cases, and culture media composition did not alter flavonoid accumulation (Figure 1C). In terms of absolute values, BD showed a significantly higher FC value, 16.0 QE, mg/g DW for MS and 15.8 QE, mg/g DW for 1/2 MS, whereas BH and BT showed similar results between 9.2 and 9.9 QE, mg/g DW. In the case of roots, FC showed a similar trend, as BD roots accumulated a significantly higher concentration of flavonoids (2.9 QE, mg/g DW) in comparison to the similar results obtained for BH and BT (Figure 1D). In addition, solvent also showed a close relationship with culture medium composition in root extracts, as similar FC results were obtained with M80 and M100 when treated with 1/2 MS.

In the case of RSA for aerial parts (Figure 1E), the results showed the same trend as observed with TPC; culture medium played a significant role, as extracts from 1/2 MS showed a significant increase in comparison to the MS counterparts (note that lower RSA values correspond to higher antioxidant activities). Additionally, M80 was the most efficient solvent in the cases of BD and BH, BH extracts causing the highest activity (0.8 mg DW mL^−1^), whereas M60 was the most efficient solvent for BT. Equally, the RSA results for root extracts showed the same behavior than TPC determination (Figure 1F) since BD root extracts showed the highest activity in both culture media tested, using M60 as solvent (2.6 mg DW mL^−1^). In contrast, RSA results for BH and BT showed a higher complexity, since it was dependent on the solvent used, with similar results from M40, M60 and M80, although an increase in RSA was reported to 1/2 MS.

According to these results, genotype, solvent and culture medium composition caused a significant effect on total phenolic content, flavonoid content and radical scavenging activity in *Bryophyllum* plants cultured in vitro. However, as unexploited plants, the characterization and identification of factors that influence phenolic compound production are two meaningful approaches that should be undertaken to assess a primary consideration on *Bryophyllum* as a valuable source of bioactive compounds. In this sense, the ANN model provides insight into the influence of each factor on all the parameters studied. As an alternative approach for multifactorial processes, ANNs require the classification of different factors into inputs, that are used by the software tool to develop the predictive model, along with the experimental results, being identified as outputs (Appendix A). Table 3 shows the results of data modeling.

All three outputs were successfully predicted by the neurofuzzy model, presenting TrainSet R^2^ values upon 70%, and the statistical assessment was proved in all cases by the higher values of *F* ratio with respect to *f* critical values (Table 3). This way, ANN models were not only able to prioritize the influence of different factors on a given variable, but they also identified the nutrient responsible for the corresponding effects, as it the case of ammonium, NH_4_^+^, among all different ions in culture media formulations (Table 1).

Additionally, the model attributed the set of “if–then” rules to all significant factors given by submodels in order to easily interpret the influence of different inputs (Table 4). To define the established levels ranged by the model as “High”, “Mid” and “Low” for every quantitative input, Form Rules^®^ also generated the corresponding values in each case (Appendix A).

For the output TPC, three submodels were generated and the strongest effect was found to be the interaction between organs and NH_4_^+^ concentration. The rules for TPC showed that it was high in aerial parts from plants grown under low NH_4_^+^ concentrations (Table 4, rule 1) and, on the contrary, TPC was low (with the strongest effect) in roots proceeding from plants grown under high NH_4_^+^ concentrations (Table 4, rule 4). These findings suggest that a reduction in NH_4_^+^ under 15 mM (Appendix A) causes an increase in the accumulation of phenolic compounds at aerial parts of in vitro-cultured *Bryophyllum* plants, whereas roots accumulate low levels with independency of culture media used. The second submodel stated that the solvent is a significant factor that impacts TPC (Table 3), and mid methanol concentrations are required to achieve a high value (Table 4, rule 6). Mid concentrations were established between 55–85% MeOH (Appendix A). Finally, a third submodel showed that the interaction between genotype and organ is a critical factor on TPC (Table 3), showing high values in aerial parts of BH (Table 4, rule 8). Altogether, the results given by the model were consistent with the findings (observed in Figure 1A,B), and data modeling conferred additional information about the significant factors that impact TPC, as discussed later.

Concerning FC, the results from the fuzzification process showed that it was dependent on the interaction of three different inputs (Table 3); solvent, organ and genotype. In this case, the FC high value with the strongest effect was obtained by aerial parts from BD (Table 4, rule 27) when using solvents with methanol proportion above 85% (Appendix A); here, the results from the model also were consistent to the previously observed (Figure 1C), and it was assessed that culture media did not cause an increase on flavonoid accumulation.

In the case of RSA, four different subcultures were reported, being the solvent factor with the highest significance (Table 3). The rules for solvent dependence showed that low values (corresponding to the highest antioxidant activity) were related to solvents with mid methanol concentrations (Table 4, rule 37), between 55–85%, as reported by the model (Appendix A). Secondary submodels showed that organ, NH_4_^+^ concentration and genotype were equally significant factors for RSA and, according to rules, extracts from aerial parts from BH and BD plants grown in low NH_4_^+^ concentrations (<15 mM; Appendix A) also enabled a higher antioxidant activity (Table 4, rules 32–41). These findings are in accordance to the previous analyses (Figure 1E,F) and, additionally, are closely related to the results from TPC modeling, thus revealing that phenolic compounds from in vitro-cultured *Bryophyllum* plants are responsible for the antioxidant activity developed by their corresponding extracts. In addition, the results from RSA of gallic acid and quercetin as shown in Figure 2, are considered as reference compounds for the major phenolic compound subfamilies found in *Bryophyllum* extracts, i.e., phenolic acids and flavonols, respectively. As can be observed, gallic acid showed an improved performance against DPPH in comparison to quercetin, thus revealing that phenolic acids could develop a more efficient activity as free-radical scavengers.

Overall, the use of ANNs conferred additional, highly valuable information about the critical factors that impact the accumulation of phenolic compounds with antioxidant activity from *Bryophyllum* in vitro-cultured plants: (1) the prioritization of determinant factors on TPC, FC and RSA, according to generated submodels; (2) the identification of deep interactions between factors; (3) the formulation of “if–then” rules for characterizing the significance and influence of each detected factor; and (4) the prediction of value intervals for determinant factors, according to experimental space (Appendix A).

## 4. Discussion

In the last few years, the interest in plant in vitro culture has been focused on medicinal plants since this technology stands out as an efficient tool to enhance the biosynthesis of secondary metabolites [21]; thus, plant in vitro culture has been successfully applied to that end, as it enables the introduction of controlled stress conditions that induce the plant defense response and leads to secondary metabolite accumulation, throughout a phenomenon called elicitation [22]. In this work, we established the in vitro culture of medicinal species belonging to *Bryophyllum* subgenus, as a solution for their low-yielding rate of bioactive compounds production. Several phytochemical reports have highlighted the need for novel approaches on the study secondary metabolism in *Bryophyllum* since large amounts of plant materials were required; as an example, for studies conducted with leaves, 0.5–20 kg of fresh material was required to test the bioactivity of *Bryophyllum* extracts [23,24,25]. On a phytochemical basis, bufadienolides have been the focus of most studies conducted on *Bryophyllum*, because of their bioactivities as anticancer and anti-inflammatory agents [9,26,27], and little attention has been paid to phenolic compounds.

Among all different stresses that could be applied under in vitro culture conditions, nutritional deficiencies are responsible for underrated abiotic stress that has a marked effect on secondary metabolite accumulation. In this sense, macronutrients play an important role in plant nutrition as they are required at high concentrations for the development of plant physiological functions. However, the dynamic range of nutrient concentrations is genotype-dependent and particularly narrow [28], so nutritional disbalances occur naturally in the plant in vitro systems. More specifically, nutritional stress has been proven to cause a strong influence on phenolic compounds levels in plant tissues; nitrogen and phosphate deficiencies have a direct impact on phenylpropanoid accumulation and potassium, sulfur and magnesium disbalances also cause an increase in phenolic compound accumulation in different plant systems [29,30,31]. Such accumulation is a consequence of the oxidative stress derived from nutrient deficiencies since this abiotic stress causes rapid changes on the cellular redox homeostasis that encompasses two further responses: (1) a genetic response, by modulating key biosynthetic genes, such as phenylalanine ammonium lyase (PAL) and chalcone synthase (CHS); and (2) a chemical response, based on the overproduction of reactive oxygen species (ROS) that stimulates the biosynthesis of antioxidant molecules to protect and maintain cell physiology [32,33].

Consequently, as unexploited medicinal plants, the establishment of *Bryophyllum* in vitro culture was performed using the universally used MS medium that allows the best plant development for most unknown plant species [4]. Thus, with the aim of increasing the production of phenolic compounds, we applied two different culture media formulations to *Bryophyllum* spp. based on this medium: full MS and half-strength macronutrient MS (1/2 MS). As stated above, the study of plant stress requires dealing with additional factors, such as genotype, organs and extraction conditions but, conversely, such studies often investigate one-by-one (one factor at time) experimental designs, that influence phenolic compound biosynthesis by omitting their interactions [34]. As a solution to this paradigm, the application of machine learning technology emerged as a solid tool to understand, predict and optimize multifactorial processes, as it has been successfully applied to other plant tissue techniques, including germination [35] and shoot multiplication [36].

For total phenolic content (TPC), three submodels were obtained and the strongest effect on this variable was developed by the interaction between the organs and NH_4_^+^ concentration (Table 3). The rules for TPC showed that it was high in aerial parts from plants grown under <15 mM NH_4_^+^ and, on the contrary, it was low (with the strongest effect) in roots proceeding from plants grown under >15 mM NH_4_^+^ concentration (Table 4). Among all different macronutrients, ammonium was spotted by the model as a critical factor related to the biosynthesis of phenolic compounds by *Bryophyllum* spp. In the case of species performing crassulacean acid metabolism (CAM), including *Bryophyllum* spp., ammonium was found to be responsible for a homeostatic disruption that leads to ROS overproduction [37]. Consequently, plant cells respond to this oxidative burst by inducing the biosynthesis of antioxidants, in which phenolic compounds are the major metabolites that exhibit such bioactivity. Roots were equally sensitive to ammonium-driven oxidative stress, and its effects may vary between different species [33]. Hence, our results suggested that ammonium played a critical role in *Bryophyllum* physiology, as an inductor of secondary metabolism. Different phytochemical screenings conducted in *Bryophyllum* spp. identified several groups of phenolics including phenolic acids, such as protocatechuic acid, ferulic acid, caffeic acid and syringic acid [11], flavonoid glycosides (see FC discussion below) and anthocyanins [38]. Additionally, a number of works have proven that phenolic biosynthesis is elicited under different stress conditions in different species of this genus [39,40].

In the case of FC, the model detected that it could be explained by one critical factor, which was the interaction between solvent, organ and genotype (Table 3). According to rules, the high values for FC were obtained, for all genotypes, in aerial parts extracted with solvents with >85% MeOH, and the strongest effect was reported for BD (Table 4). These results indicate that flavonoids synthesized by *Bryophyllum* spp. were more likely soluble in solvents ranging from 85–100% MeOH. An important factor that impacts phenol solubility, including flavonoids, is pH [41], but as we did not alter the pH of the extracts, this factor cannot be analyzed according to our results. Additionally, the higher accumulation of flavonoids in aerial parts spotted by the model could be explained from a physiological point of view, since these compounds promote effective protection against UV-radiation and they tend to accumulate in leaves and flowers, as reported in this genus [42]. The studies focused on flavonoid identification performed in *Bryophyllum* spp. showed that species from this subgenus synthesize mainly flavonol glycosides, such as quercetin and kaempferol glycosides, and flavone glycosides [23,25]. Recently, it has been reported that flavonoid biosynthesis can be elicited under cyclodextrin-mediated stress in plant suspension-cultured cells from BH [40].

Concerning RSA, the model identified that the major factor that could explain the variability for this output was the solvent (Table 3). In order to better understand the rules obtained for RSA, a lower IC50 value indicates a higher antioxidant activity; consequently, the lowest value for RSA was reported when the solvent was mid (Table 4), ranging from 55–85% MeOH. It is noteworthy that these results responded to only one antioxidant process, as it was radical scavenging activity against the free-radical DPPH. However, the antioxidant activity involves multiple components, such as redox, hydrogen-donating and chelating processes [43]. The use of DPPH constitutes a fast and reliable method for monitoring the scavenging of free-radicals by plant extracts containing antioxidant compounds. It offers highly valuable information for the valorization of the health-enhancing properties of plant extracts, as free-radicals are responsible for many deleterious cellular processes, including oxidative stress, aging and carcinogenesis [8].

With respect to phenolic content, low RSA values were obtained with mid methanol concentrations in the solvent, which corresponded to high TPC values, as seen for submodel 2 (Table 4). In general, phenols possessed a higher affinity towards solvents with mid-solubility, such as organic alcohols and acetone. However, phenolic compounds showed a preference for alcohols, mainly methanol and ethanol, since their hydroxylated moiety enabled the development of hydrogen bonds with the oxygen atoms included on phenolic structures [44]. Furthermore, due to the great phenolic heterogeneity, the addition of variable water fractions to these solvents promoted a higher extraction efficiency of phenolic compounds that were soluble in water and organic solvents at the same time [45]. These findings indicated that the solvent required for increased extraction of phenolic compounds also promoted a high radical scavenging activity using 55–85% MeOH as solvents. However, high RSA values were found at high methanol concentrations in the solvents (Table 4), as was the case of high flavonoid concentrations. This fact could be explained according to two different hypotheses: (1) DPPH is a free-radical that exhibits a high chemical stability in methanol, so the use of methanolic extracts above 85% MeOH may improve such stability, although antioxidants are present in the extract [46]; (2) glycosylation promotes the inhibition of antioxidant activity of many flavonols, with respect to their free counterparts, and glycosylated flavonols are the major flavonoids found in *Bryophyllum* spp. [25,47]. As a consequence, the inclusion of an initial acid hydrolysis step during phenolic extraction may improve the antioxidant activity of flavonoids, as it enables the production of free flavonols [48]. Equally, both hypotheses are in line with results obtained from RSA and from reference compounds (gallic acid and quercetin), where quercetin showed a lower efficiency against DPPH inhibition (Figure 2). This observation could be a consequence of the putative activity of flavonoids as antioxidants, being effective agents for metal chelation, and maintaining redox homeostasis, better than free-radical scavengers [43].

The information obtained from neurofuzzy logic data modeling enabled a solid, easier interpretation of results and, additionally, the derived data can be used for further studies in order to optimize the production of phenolic compounds by in vitro-cultured *Bryophyllum* plants. As it was reported, a 50% reduction on macronutrient concentration in the culture media was accompanied by a rise in the total phenolic content of *Bryophyllum* extracts; future reports will be applied in this sense, by increasing the experimental space with the aim of getting a more feasible perspective of mineral nutrition in these species and their impact on secondary metabolism. In parallel, further studies will also focus on the interaction of macronutrients and micronutrient deficiencies, as they also have been reported as modulators of phenolic compounds biosynthesis.

## 5. Conclusions

In this work, we combined, for the first time, the establishment of plant in vitro culture with neurofuzzy logic with the aim of characterizing and optimizing the experimental conditions for the production of phenolic compounds by *Bryophyllum* spp. under nutritional stress. Throughout the application of machine learning methodology, the proper algorithms were able to learn from experimental observations and build up a model with prediction capabilities in order to characterize the three variables used for this work: total phenolic content, flavonoid content and radical-scavenging activity. Our results suggest that the maximum yield of phenolic compounds was reported when using aerial parts from BH cultured in 1/2 MS medium as a source of phenolic compounds. The model identified ammonium concentration as the nutritional factor that influences the biosynthesis of phenolic compounds by *Bryophyllum* spp. The maximum antioxidant activity was achieved using aqueous methanol (55–85% MeOH) as a solvent, which was the most efficient solvent for the extraction of total phenolics. On the other hand, flavonoids were better extracted with solvents with a higher methanol proportion (>85% MeOH), and their biosynthesis was independent of the culture medium composition. Finally, we proposed the combination of two cutting-edge methodologies, plant in vitro culture and artificial intelligence-based tools, to achieve a primary approach to the phytochemical potential of unexploited medicinal plants.

## Figures and Tables

**Figure 1 antioxidants-09-00210-f001:**
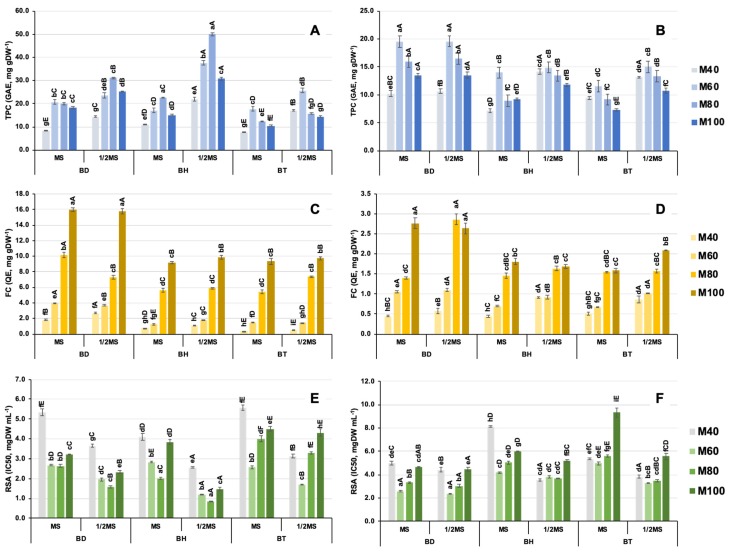
Results from different determinations of phenolic compounds and antioxidant activity. (**A**) TPC in aerial parts, (**B**) TPC in roots, (**C**) FC in aerial parts, (**D**) FC in roots, (**E**) RSA in aerial parts, and (**F**) RSA in roots. Values represent the mean of three independent extracts, and vertical bars represent standard deviation. Lower case letters (a–d) indicate significant differences between solvents and genotypes for the same culture medium (*p* < 0.05), and capital letters (A–D) indicate significant differences between culture media and genotypes for the same solvent (*p* < 0.05).

**Figure 2 antioxidants-09-00210-f002:**
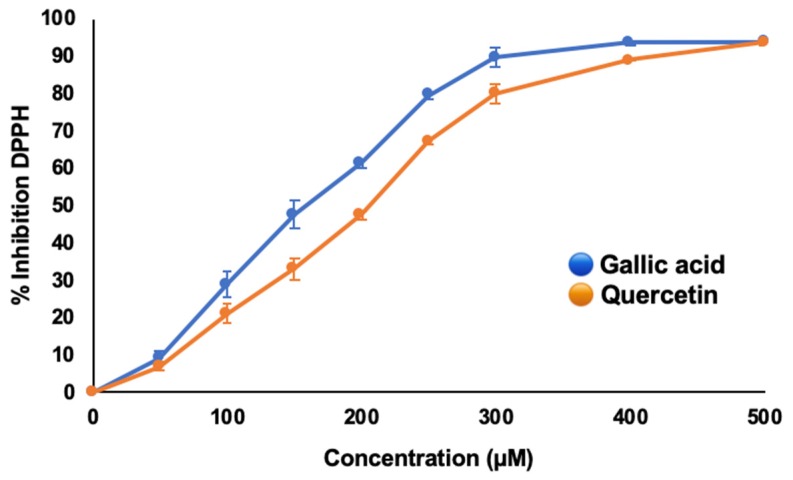
Results for RSA of reference compounds: gallic acid for phenolic acids and quercetin for flavonols. Results are expressed as the mean of the inhibition percentage of DPPH from three independent replicates, and vertical bars indicate standard deviation.

**Table 1 antioxidants-09-00210-t001:** Ion contents obtained by the decomposition of macronutrient salts used for culture media tested in this work.

Ions	MS (mM)	1/2 MS (mM)
NO_3_^−^	39.4	19.7
NH_4_^+^	20.6	10.3
K^+^	20.0	10.0
Cl^−^	5.99	2.99
Ca^2+^	2.99	1.50
Mg^2+^	1.50	0.75
HPO_4_^2−^	1.25	0.62
SO_4_^2−^	1.76	1.01

Hence, a total of 11 factors were selected as inputs: genotype, organ, 8 ions (Table 1) and solvent used for phenolic extraction. On the other hand, three parameters were included as outputs: total phenolic content (TPC), flavonoid content (FC) and radical-scavenging activity (RSA; Appendix A). MS stands for Murashige and Skoog medium.

**Table 2 antioxidants-09-00210-t002:** Training parameters used by FormRules v4.03 for model construction.

Minimization Parameters
Ridge regression factor: 1 × 10^−6^
MODEL SELECTION CRITERIA
Structural risk minimization (SRM)
C1 ≥ 0.85 C2 = 4.8
Number of set densities: 2
Set densities: 2, 3
Adapt nodes: TRUE
Max. inputs per submodel: 4
Max. nodes per input: 15

**Table 3 antioxidants-09-00210-t003:** Critical factors for each output and quality parameters of neurofuzzy logic models.

Outputs	Submodel	Train Set *R*^2^	*F* ratio	df1, df2	*f* Critical (α < 0.05)	Significant Inputs
TPC	**1**	75.75	10.22	11, 47	2.00	**Organ × NH_4_^+^**
	2					Solvent
	3					Genotype × Organ
FC	**1**	98.10	83.23	18, 47	1.83	**Genotype × Organ × Solvent**
RSA	1	72.33	14.94	7, 47	2.21	Organ
	2					NH_4_^+^
	**3**					**Solvent**
	4					Genotype

Bold inputs correspond to the stronger effect for each output.

**Table 4 antioxidants-09-00210-t004:** Rules selection obtained by neurofuzzy logic.

Rules		Gen.	Org. ^1^	Solv. ^2^	NH_4_^+^		TPC	FC	RSA	Membership
1	IF		**A**		**Low**	THEN	**High**			**0.75**
2			A		High		Low			0.99
3			R		Low		Low			0.85
4			**R**		**High**		**Low**			**1.00**
5				Low			Low			1.00
6				Mid			High			0.70
7				High			Low			0.83
8		BH	A				High			0.63
9		BH	R				Low			1.00
10		BD	A				Low			0.76
11		BD	R				Low			0.80
12		BT	A				Low			1.00
13		BT	R				Low			1.00
14	IF	BH	A	Low		THEN		Low		0.98
15		BD	A	Low				Low		0.88
16		**BT**	**A**	**Low**				**Low**		**1.00**
17		BH	R	Low				Low		0.98
18		BD	R	Low				Low		0.99
19		BT	R	Low				Low		0.98
20		BH	A	Mid				Low		0.84
21		BD	A	Mid				Low		0.71
22		BT	A	Mid				Low		0.80
23		BH	R	Mid				Low		0.95
24		BD	R	Mid				Low		0.92
25		BT	R	Mid				Low		0.95
26		BH	A	High				High		0.60
27		**BD**	**A**	**High**				**High**		**1.00**
28		BT	A	High				High		0.60
29		BH	R	High				Low		0.91
30		BD	R	High				Low		0.85
31		BT	R	High				Low		0.90
32	IF		A			THEN			Low	0.97
33			R						High	0.81
34					Low				Low	0.90
35					High				High	0.74
36				Low					High	0.61
37				**Mid**					**Low**	**1.00**
38				**High**					**High**	**0.67**
39		BH							Low	0.79
40		BD							Low	0.94
41		BT							High	0.57

^1^ “A” refers to aerial parts, and “R” refers to roots. ^2^ Solvent was expressed as methanol proportion within the solvent. Bold letters indicate inputs with the strongest effect on each output, as indicated by the model. “Gen.” refers to genotype; “Org.” refers to organ; “Solv.” refers to solvent.

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
