# Peer review of "Combining Medicinal Plant In Vitro Culture with Machine Learning Technologies for Maximizing the Production of Phenolic Compounds"

_antioxidants, 2020, doi:10.3390/antiox9030210_

Round 1
Reviewer 1 Report
The paper entitled: "Combining medicinal plant in vitro culture with machine learning technologies for maximizing the production of phenolic compounds." by Pascual García-Pérez et al is a very interesting manuscript presenting a new approach provoding a new approach of the study of natural products.
Major Comments
- I suggest to provide a deeper analyis on the correlation of these results with other pure compounds (used as positive pcontrols)
- I suggest to provide more information regarding DPPH test a]
Reviewer 2 Report
Suggestions:
- Lines 166-168 (Formula 1): write on one line
- Line 169: "yi stand" - I suggest "where yi stand"
- Line 169: replace "yi stand" with "where yi stand"
- Line 170: replace "anti-radical" with "radical"
- In Table 3: "TPC-" replace with "TPC"
- In Table 3: "RSA)" replace with "RSA"
